# The Diagnostic and Therapeutic Role of snoRNA and lincRNA in Bladder Cancer

**DOI:** 10.3390/cancers15041007

**Published:** 2023-02-04

**Authors:** Hao Wang, Yanfei Feng, Xiangyi Zheng, Xin Xu

**Affiliations:** 1Department of Urology, First Affiliated Hospital, School of Medicine, Zhejiang University, Hangzhou 310003, China; 2Second Affiliated Hospital, School of Medicine, Zhejiang University, Hangzhou 310003, China

**Keywords:** bladder cancer, small nucleolar RNA, long intergenic noncoding RNA

## Abstract

**Simple Summary:**

Bladder cancer is one of the most common urological malignancies. Despite the continuous improvement of diagnosis and treatment in recent years, the high recurrence rate and poor prognosis of bladder cancer often led to poor quality of life. In the past few years, emerging evidence has demonstrated that long noncoding RNAs play a crucial role in the carcinogenesis and progression of bladder cancer. Small nucleolar RNAs (snoRNAs) and long intergenic noncoding RNAs (lincRNAs) are both subclasses of lncRNAs, and their dysregulation has been proven to be closely related to tumour growth and metastasis. In this review, we focus on the recently published literature on snoRNAs and lincRNAs in bladder cancer and summarize the important role of snoRNA and lincRNA in the carcinogenesis and development of bladder cancer. Our work contributes to a better understanding of the specific mechanism of the role of snoRNA and lincRNA in bladder cancer and reveals their potential as diagnostic biomarkers and therapeutic targets.

**Abstract:**

Bladder cancer is one of the most common malignancies of the urinary tract and can be divided into non-muscle-invasive bladder cancer (NMIBC) and muscle-invasive bladder cancer (MIBC). Although the means of diagnosis and treatment have continually improved in recent years, the recurrence rate of bladder cancer remains high, and patients with MIBC typically have an unfavourable prognosis and a low quality of life. Emerging evidence demonstrates that long noncoding RNAs play a crucial role in the carcinogenesis and progression of bladder cancer. Long intergenic noncoding RNAs (lincRNAs) are a subgroup of long noncoding RNAs (lncRNAs) that do not overlap protein-coding genes. The potential role of lincRNAs in the regulation of gene expression has been explored in depth in recent years. Small nucleolar RNAs (snoRNAs) are a class of noncoding RNAs (ncRNAs) that mainly exist in the nucleolus, are approximately 60–300 nucleotides in length, and are hosted inside the introns of genes. Small nucleolar RNA host genes (SNHGs) have been associated with the origin and development of bladder cancer. In this review, we aim to comprehensively summarize the biological functions of these molecules in bladder cancer.

## 1. Introduction

Bladder cancer (BLCA) is the fifth most common malignancy and the second most common urological malignancy globally. With increasing incidence and mortality rates, BLCA was estimated to account for 81,180 new cases and 17,100 deaths in 2022 [1]. Approximately 70% of patients are diagnosed with non-muscle-invasive bladder cancer (NMIBC), most of whom are treated with transurethral resection of the bladder tumour (TURBT) and have a 5-year survival rate greater than 85%. The other 30% of patients are diagnosed with muscle-invasive BC (MIBC), need to receive comprehensive treatment, and have a low 5-year survival rate [2]. In addition, previous studies have shown that the recurrence rate of bladder cancer is extremely high. Almost 70% of high-risk NMIBC cases recur in 10 years, and diagnosis and surveillance of bladder cancer still rely on repeated cystoscopic biopsy. Therefore, exploring novel therapeutic targets and potential biomarkers of BLCA is essential [3,4,5].

Non-protein coding genes, which account for approximately 98% of the human genome, were long regarded as useless genes [6]. However, with the development of next-generation sequencing technology, researchers have started to turn their attention to this unexplored area. Among ncRNAs, transport RNA (tRNA), ribosomal RNA (rRNA) and messenger RNA (mRNA) are well known for being widely involved in cellular processes. The remaining ncRNAs account for a small proportion with diverse classes, including small nuclear RNA (snRNA), small nucleolar RNA (snoRNA), microRNA (miRNA), PiWi-interacting RNA (piRNA) and long noncoding RNA (lncRNA) [7,8]. Since ncRNAs were put into the spotlight, researchers have found that the gain and loss of these nonprotein coding genes regulate the progression of various diseases, especially cancers.

SnoRNAs are small RNAs with a length of 60–300 nucleotides that are predominantly located on the nucleolus. SnoRNAs belong to the following two major families according to secondary structure and signature sequence: the C/D box snoRNA family and the H/ACA box snoRNA family. The major function of SnoRNAs is acting as guide RNAs for posttranscriptional modification of rRNA, tRNA and snRNAs. However, in the past few years, an increasing number of studies have demonstrated that the dysregulation of snoRNAs is also involved in regulating oncogenesis and cellular processes [9,10,11,12]. In humans, snoRNAs are mainly transcribed from host genes; snoRNA host genes (SNHGs), which contain both introns and exons in their sequences, play a primary role in regulating the expression of snoRNAs. During transcription, introns in SNHGs are removed, processed and matured into snoRNA and function mainly in the nucleolus [9,13].

The lncRNA family (>200 nt in length) is a large and multifunctional family. Most lncRNAs are located in the nucleus. LncRNAs can directly interact with DNA, therefore regulating gene transcription or influencing RNA splicing through binding with RNA binding proteins. In the cytoplasm, lncRNAs are also involved in multiple processes, including translation, RNA localization and posttranslational modification [6,14]. To date, the function of most lncRNAs has not been identified; thus, the classification of lncRNAs relies on their location with respect to protein-coding genes, which are categorized as antisense, overlapping, intronic, host genes, intergenic lncRNAs (lincRNA) and enhancer RNAs. LincRNA is a subgroup of lncRNA and is transcribed from protein noncoding genomic regions [15]. Similar to other lncRNAs, most lincRNAs are located in the nucleus. Recent studies have revealed that lincRNAs also contribute to tumour progression through various mechanisms, including directly regulating chromatin stability, scaffolding specific proteins and sequestering miRNAs [16,17].

In this review, we aim to comprehensively summarize the biological functions of important snoRNAs and lincRNAs in bladder cancer.

## 2. Pathophysiology of BLCA

The most common pathological type of BLCA is bladder urothelial carcinoma. According to the depth of tumour invasion to bladder mucosa, BLCA can be divided into NMIBC and MIBC. Specifically, tumours at the Ta stage, T1 stage and carcinoma in situ (CIS) belong to NMIBC, and tumours at T3 and T4 stages belong to MIBC. These two types represent two periods of occurrence and development of BLCA, and their treatment methods and prognosis of them are quite different. TURBT and intravesical chemotherapy are the main treatment methods of the former, while the latter requires radical cystectomy or systemic treatment. Recently, in the published literature indicated 8 SNHGs (SNHG1, SNHG2, SNHG5, SNHG7, SNHG12-14 and SNHG20) and 6 lincRNAs (linc0001, linc00023, linc00047, linc00080, linc00178 and linc00319) that correlate with the TNM stage of BLCA (Figure 1).

We downloaded RNA-seq data from The Cancer Genome Atlas bladder cancer (TCGA-BLCA) dataset. Then “limma” package was used to explore necroptosis-related differentially expressed genes (DEGs) between normal and tumour samples under a false discovery rate (FDR) < 0.05 and |log2fold change (FC) >1|. SNHG14, linc00023 and linc00641 were high expressed in BLCA samples, while SNHG1, SNHG3, SNHG4, SNHG12, SMHG209, linc00355, linc00649, linc00958 and linc00178 were expressed at low levels (Figure 2).

## 3. SnoRNA and lincRNA

SnoRNAs are highly conserved RNAs that can be traced back to archaea and eukaryotes 20–30 billion years ago. They are mainly localized in the nucleus and have a length of 60–300 nt [18]. The main functions of snoRNAs include guiding methylation and pseudouridylation of rRNA, alternative splicing of mRNA, telomere synthesis and other unknown cell processes [6]. There are the following two main families of snoRNAs according to their secondary structure and signature sequence: C/D boxes and H/ACA boxes. All snoRNAs from the two families need to bind to highly conserved core proteins to form snoRNPs and function in vivo. In the C/D box family, the core proteins NOP5, NOP5 and NHPX form a bridge that links C/D guide RNA and the methyltransferase fibrillarin. The C/D RNP guides 2′-O-ribose methylation. In the H/ACA box family, RNA can directly interact with the pseudouridine synthase dyskerin and three core proteins, GAR1, NOP10 and NHP2, forming H/ACA BNP and guiding pseudouridylation [9,10,19]. In humans, snoRNAs are generated from host genes, especially SNHGs. SNHGs can act in several ways with different subcellular compartment localizations. Inside the nucleus, it can directly interact with transcription factors and methylating enzymes to modulate gene transcription. In the cytoplasm, it can sponge miRNA, directly bind to mRNA or interact with proteins to regulate translation [13] (Figure 3).

LincRNAs are transcribed between two protein-coding genes with a length of more than 200 nt. Although lincRNAs do not code proteins directly, they play an important role in multiple cellular processes like other ncRNAs. Most lincRNAs are located in the nucleus rather than the cytoplasm and may impact malignant progression in various ways. Several lincRNAs can control gene expression by regulating chromatin in cis and in trans. In the nucleus, lincRNAs could not only facilitate the bind between DNA methyltransferases (DNMTs) and gene promoters, inhibiting the expression of specific genes but also directly silence tumour suppressor genes. Moreover, a number of lincRNAs sponge miRNAs that usually interact with their target gene at the 3′ untranslated region and form silencing complexes, thus enhancing the expression level of target mRNAs [16,17,20] (Figure 3).

## 4. The Role of snoRNA in BLCA

SnoRNAs have been identified to participate in the progression of multiple cancers, including endometrial cancer, hepatocellular carcinoma and BLCA. The latest studies on the functions and mechanisms of SNHGs in BLCA are described below (Table 1).

SNHG1, which is located at chromosome 11q12.3, is overexpressed in several types of tumours, including pancreatic, prostate, non-small-cell lung and BLCA [21,22,23,24]. It is widely considered an oncogene that promotes cancer proliferation, migration, invasion and tumorigenesis. SNHG1 can bind and coregulate with the PP2A catalytic subunit (PP2A-c) to promote c-Jun phosphorylation. Then, activated c-Jun increases matrix metalloproteinase 2 (MMP2) transcription, which induces cancer cell invasion and metastasis. Additionally, transcription of miR-34a was downregulated by SNHG1 via autophagy, thus maintaining MM2 mRNA stabilization [21]. A recent study revealed that elevated SNHG1 could interact with the DNA methyltransferase DNMT3A and then bind to and hypermethylate the promoter of miR-129-5p. The low expression of miR-129-5p maintains the stability of Rac1 mRNA, resulting in increased stemness and invasiveness of BLCA cells [22]. Moreover, SNHG1 upregulates the histone methyltransferase EZH2 by direct interaction and functions as a sponge of miR-137-3p. Overexpressed EZH2 silenced CDH1 (also known as E-cadherin) and KLF2 and led to the metastasis of BLCA cells [23,24]. Cai et al. found that SNHG1 could also increase BLCA cell proliferation through the SNHG1/miR-9-3p/MDM2/PPARγ axis [25].

SNHG2, which is also known as growth arrest-specific transcript 5 (GAS-5), is located at chromosome 1q25 and acts as a tumour suppressor in multiple cancers [26,27,28,29]. Studies have found that SNHG2 is expressed at low levels in BLCA cells and tissues and that the expression level of SNHG2 is markedly correlated with the clinical characteristics and prognosis of BLCA. High expression of SNHG2 promotes BLCA cell apoptosis and inhibits tumour proliferation. SNHG2 directly interacts with E2F transcription factor 4 (E2F4) and recruits it to the enhancer of the zeste homolog 2 (EZH2) promoter, thereby downregulating transcription of the EZH2 oncogene [26]. In the BLCA cell line HTB-9, SNHG2 sponges miR-21, thus promoting transcription of its downstream gene phosphatase and tensin homolog (PTEN), leading to the downregulation of antiapoptotic proteins and cell cycle-associated proteins, which suppresses proliferation and increases apoptosis of bladder cancer cells [27]. According to previous studies, downregulation of SNHG2 could also directly arrest the cell cycle at the S phase in a cyclin-dependent kinase 6 (CDK6)-dependent manner [28]. In the doxorubicin-resistant BLCA cell line T24, the level of SNHG2 was low. However, overexpression of SNHG2 could abolish chemotherapy resistance by promoting doxorubicin-induced apoptosis. This could be realized by upregulating the Bcl-2 expression [29]. Additionally, the knockdown of SNHG2 could also reverse the growth inhibition of BLCA cells induced by gambogic acid, a promising anticancer compound [26].

SNHG3 is also located at chromosome 1 but is generally considered an oncogene. Studies have demonstrated that SNHG3 is upregulated in BLCA cells and tissue and is positively correlated with poor clinicopathological characteristics and prognosis [30]. Overexpression of SNHG3 in the BLCA cell line facilitated cell growth, metastasis and tumorigenesis through the SNHG3/c-MYC/BMI1 axis. The knockdown of SNHG3 significantly suppressed the proliferation of BLCA both in vivo and in vitro. Interestingly, the knockdown of SNHG3 barely induced lower transcription of c-MYC in vivo [31]. Furthermore, SNHG3 also promotes the expression of Go-Ichi-Ni-San2 (GINS2), a subunit of the cell cycle regulating complex GINS, by acting as a sponge of miR-515-5p, thus promoting tumour proliferation and epithelial-mesenchymal transition (EMT) [32].

SNHG5 and SNHG6 are oncogenes located at 6q14.3 and 8q13.1, respectively. The dysregulation of these two genes was found to be closely related to tumour growth and metastasis in multiple cancers [33,34]. Nevertheless, there is still a lack of research on the functions of SNHG5 and SNHG6 in bladder cancer. Ma et al. noted that SNHG5 was upregulated in both BLCA tissues and cell lines, and higher expression of SNHG5 was markedly correlated with poorer tumour clinicopathological features and overall survival (OS). The authors also found that silenced SNHG5 attenuated cell proliferation by arresting the cell cycle at the G1 phase in a p27-dependent manner [35]. Similarly, overexpression of SNHG6 increased the migratory and invasive abilities of BLCA cells by competitively binding to miR-125b, thus enhancing the expression of Snail1/2 and NUAK family kinase 1 (NUAK1) [34].

SNHG7 and SNHG20 are novel host genes located at chromosome 9q34.3 and 17q25.2, respectively. Upregulation of SNHG7 and SNHG20 has been connected to the progression of several cancers, including BLCA, and linked to increased invasive capacity along with poor OS [36,37]. Proliferation, proapoptotic and EMT-related protein levels were regulated when SNHG7 was knocked down. The levels of PCNA and Ki-67 (proliferation-related), Bax and cleaved caspase 3 (proapoptotic-related) were elevated, while MMP2, MMP7 and E-cadherin (EMT-related) were reduced [38]. In addition, the inactivation of the Wnt/β-catenin signalling pathway and activation of the Src/FAK signalling pathway also played crucial roles in promoting the growth, migration and invasion of BLCA cells. Finally, SNHG7 could also be activated by the oncogene ELK1 and function as a molecular sponge of miR-2682-5p, which suppresses the transcription of ELK1, thus forming a positive feedback loop [38,39]. SNHG20 also contributes to the migration and invasion of BLCA cells via the Wnt/β-catenin signalling pathway [37].

SNHG13 (also known as DANCR) is located at chromosome 4q12 and was first identified in primary human keratinocytes in 2012 [40]. Recent studies have found that SNHG13 is aberrantly upregulated in BLCA tissues and cells and accounts for the proliferation and metastasis of BLCA cells. In the cytoplasm, SNHG13 can directly interact with leucine-rich pentatricopeptide repeat containing (LRPPRC) and stabilize interleukin 11 (IL-11), plasminogen activator urokinase (PLAU) and CCND1 mRNAs. The enhanced expression of IL-11 led to the activation of the IL-11/JAK/STAT3 axis and drove the proliferation, invasion and migration of BLCA cells. PLAU and CCND1 play important roles in tumour cell migration and the cell cycle, respectively [41]. Additionally, SNHG13 can also regulate Musashi RNA binding protein 2 (MSI2) and vascular endothelial growth factor C (VEGF-C) to influence cell motility in a ceRNA-dependent manner by acting as sponges of miR-149 and miR-335, respectively [42,43].

Apart from the SNHGs mentioned above, SNHG12, 14, 15 and 16 are all elevated in BLCA cells and are positively correlated with poor prognosis. SNHGs in tumours contribute to the proliferation, migration and invasion of tumour cells [44,45,46,47,48,49]. Among them, SNHG14 and 16 were proven to function as competing for endogenous RNAs (ceRNAs). SNHG14 sponges miR-150-5p and miR-211-3p, while SNHG16 sponges miR-200a-3p [45,46,48]. In addition, SNHG16 can recruit EZH2 to the promoter of p21, thus silencing p21 at the transcriptional level [49].

**Table 1 cancers-15-01007-t001:** The role of snoRNA SNHGs in bladder cancer.

Name	Expression	Related miRNAs, Proteins or Pathways	Related Tumour Cell Biology	Relevant Clinical Features	Reference
SNHG1	Upregulated	miR-34a, miR-129-2-5p, miR-143-3p, miR-137-3p, miR-9-3p	Proliferation, migration, invasion, apoptosis, autophagy, EMT and stemness	TNM stage, LN invasion, metastasis, recurrence-free survival and prognosis	[21,22,23,24,25]
SNHG2 (GAS-5)	Downregulated	miR-101, miR-21, CDK6	Proliferation, migration, apoptosis and doxorubicin resistance	Age, tumour size, tumour stage, LN invasion and prognosis	[26,27,28,29]
SNHG3	Upregulated	miR-515-5p, c-MYC	Proliferation, migration, invasion, EMT and angiogenesis	TNM stage and prognosis	[30,31,32]
SNHG5	Upregulated	p27	Migration and apoptosis	Stage, tumour size, LN invasion, metastasis and prognosis	[35]
SNHG6	Upregulated	hsa-miR-125b	Invasion and migration	-	[34]
SNHG7	Upregulated	miR-2682-5p, Wnt/β-catenin	Proliferation, invasion and migration	Stage, tumour size, LN invasion and prognosis	[37,38,39]
SNHG12	Upregulated	-	Proliferation	-	[44]
SNHG13 (DANCR)	Upregulated	miR-335, miR-149, IL-11	Proliferation, invasion, migration and lymphatic metastasis	TNM stage, histological grade, LN invasion and prognosis	[41,42,43]
SNHG14	Upregulated	miR-150-5p, miR-211-3p	Proliferation, invasion, migration and apoptosis	TNM stage, LN invasion and prognosis	[45,46]
SNHG15	Upregulated	-	Proliferation and invasion	-	[47]
SNHG16	Upregulated	EMT	Invasion and migration	-	[48]
SNHG20	Upregulated	Wnt/β-catenin	Proliferation, invasion, migration and apoptosis	Stage, LN invasion and prognosis	[37]

Abbreviations: SNHG, small nucleolar RNA host genes; EMT, epithelial-mesenchymal transition; LN, lymph node; CDK6, cyclin-dependent kinase 6; c-MYC, MYC proto-oncogene.

## 5. The Role of lincRNA in BLCA

An increasing number of lincRNAs have been shown to be differentially expressed between bladder cancer and normal tissues. Further studies have demonstrated that some of them are associated with the tumorigenesis and progression of bladder cancer. We describe these valuable studies below (Table 2).

Linc00001, also known as X-inactive specific transcript (XIST), is located on chromosome Xq13.2. It was first found in the process of X chromosome inactivation, which occurs at the early development stage of mammalian females, and was later shown to act as an oncogene and to be upregulated in multiple types of cancers, including BLCA [50]. Recent studies have revealed that XIST can regulate cancer cell progression by acting as a sponge of miRNAs and binding to proteins [51,52,53,54]. Specifically, the expression of XIST was elevated in BLCA tissues and cell lines, including T24, 253J, RT112 and HT-1376. XIST silencing induced the loss of proliferation, metastasis and stemness capability combined with the overexpression of miR-200c and miR-133a in vitro [53,54]. In addition, Chen et al. also found that the downregulation of XIST is probably one of the mechanisms of the antitumour drug platycodon D (PD). They noticed that PD-treated BLCA cells had poor proliferation, migration and invasion capabilities in vivo and in vitro; this was associated with low expression levels of XIST and high expression levels of miR-355. Silencing XIST or overexpressing miR-355 aggravated the antitumour effects of PD in vitro, indicating that XIST/miR-355 are related to sensitivity to the antitumour drug PD [51]. Another study determined that XIST was mainly distributed in the nucleus and focused on its potential regulatory role at the transcriptional level. They found that XIST downregulated the expression level of p53 by interacting with tet methylcytosine dioxygenase 1 (TET1) and then binding to the promoter of p53 [52].

Linc00023 is located on chromosome 14q32.2 within the DLK-MEG3 locus and is also widely known as the maternally expressed 3 gene (MEG3). The deregulation of linc00023 was reported to be involved in the progression of various tumours, including meningioma, hepatocellular cancer, breast cancer and BLCA [55,56]. Similarly, MEG3 can also act as a ceRNA to regulate tumour cell proliferation and apoptosis. In bladder cancer, MEG3 was found to function as a ceRNA for PTEN by competitively binding miR-494, thus repressing the proliferation, migration and invasion and promoting the apoptosis of tumour cells [57]. In addition, the MEG3/miR-96/Tropomyosin (TPM) axis functions in the same manner. Mir-96 was sponged by MEG3 and led to the increased expression level of TPM, thus inhibiting tumour progression in vivo and in vitro [58]. Interestingly, overexpression of MEG3 in BLCA cells also enhanced chemosensitivity to cisplatin and expression of p53 [59]. MEG3 also upregulated the PH domain and leucine-rich repeat protein phosphatase 2 (PHLPP2) by sponging miR-27a. PHLPP2 could directly interact with and suppress the phosphorylation of c-JUN, thus downregulating the transcription of c-MYC and ultimately restraining tumour metastasis [60].

Linc00047 is also known as metastasis-associated lung adenocarcinoma transcript 1 (MALAT1) and has a length of 8.5 kb. It is located on chromosome 11q13, is involved in the regulation of several molecular signalling pathways and has been shown to be a potential biomarker in bladder cancer, nasopharyngeal carcinoma and osteosarcoma [61,62]. The expression of MALAT1 was significantly elevated in BLCA tissues and cell lines and was positively correlated with advanced clinical stage and poor prognosis [63]. In addition, MALAT1 sponged miR-101-3p and suppressed the expression of VEGF-C in BLCA 5637 and EJ-M3 cell lines, enhancing cisplatin sensitivity. Furthermore, the knockdown of MALAT1 in cisplatin-resistant 5637 and EJ-M3 cells could reverse drug resistance [64]. Tao et al. revealed that fat mass and obesity-associated protein (FTO), an RNA m6A demethylase, promotes MALAT1 expression by restraining its m6A modification. Then, MALAT1 acts as a sponge for miR-384, thus enhancing the transcription level of oncogene mal T-cell differentiation protein 2 (MAL2) and ultimately promoting bladder cancer growth. In accordance with previous findings, the clinical expression levels of FTO, MALAT, miR-384 and MAL2 were collected; the results also showed a strong correlation between them [65].

Linc00080, also known as taurine-upregulated gene 1 (TUG1), is located on chromosome 22q12 and is almost twice as elevated in bladder tissues compared to adjacent normal tissues. It is broadly considered an oncogene in several tumours, such as colorectal cancer, oesophageal cancer, osteosarcoma and bladder cancer [66]. In BLCA, TUG1 is involved in tumour proliferation, metastasis and apoptosis, as well as radioresistance and chemotherapy resistance [67,68,69,70,71]. Silencing TUG1 in BLCA cell lines restrains the tendency of EMT by acting as a ceRNA of ZEB2 by sponging miR-145 and leading to radioresistance [67]. It also promotes the expression of high mobility group box 1 protein (HMGB1), thus enhancing bladder cancer radioresistance in vivo and in vitro, yet the mechanism needs further exploration [68]. Similarly, TUG1 also enhanced proliferation and metastasis and cisplatin resistance and induced apoptosis of BLCA cells by acting as sponges of miR-194-5p and miR-140-3p [69,70]. In addition, in the BLCA cell lines T24 and BIU-87, overexpressed TUG1 competitively binds with miR-142, thus stabilizing zinc finger E-box binding homeobox 2 (ZEB2) and upregulating the transcription of the Wnt/β-catenin pathway proteins β-catenin, cyclin D1 and c-Myc [71].

Linc00178 is located on 19p13.12 and was first identified in bladder transitional cell carcinoma; thus, linc00178 is also known as urothelial cancer associated 1 (UCA1). Recent studies have verified the overexpression and oncogene functions of UCA1 in different tumours [72]. Specifically, UCA1 can serve as a sponge of miRNAs to regulate tumour growth, metastasis, drug resistance and mitochondrial functions. Li et al. found that knockdown of UCA1 in the BLCA cell line 5637 decreased mitochondrial DNA copy number by more than half with accompanying decreased ATP production. The results of a UCA1 overexpression experiment on the BLCA cell line UMUC2 were highly consistent with those of the knockdown experiments. Further studies revealed that UCA1 could sponge mIR-195-5p, thus elevating the level of the miR-195-5p target gene ADP-ribosylation factor-like 2 (ARL2), which plays a crucial role in the transport of ATP/ADP [73]. Second, UCA1 also sponges miR-582-5p and miR-143 and contributes to proliferation, invasion and drug resistance through ATG7-mediated autophagy and HMGB1-mediated EMT [74,75]. Furthermore, Xue et al. demonstrated that hypoxia enhanced the transfer of UCA1 by stimulating small extracellular vesicle (exosome) secretion, and in turn, UCA1 reshaped the tumour microenvironment and promoted tumour progression [76].

Linc00319 is a novel lncRNA located on chromosome 21q22.3 that has been implicated in the tumorigenesis and progression of cervical cancer, gastric cancer, osteosarcoma and laryngeal squamous cell carcinoma in a miRNA-dependent manner [77,78,79,80]. Moreover, linc00319 expression levels were remarkably higher in BLCA tissues than in adjacent normal tissues. Patients with higher linc00319 levels had higher clinical stages and lower recurrence-free survival rates [81]. In addition, in the BLCA cell lines T24 and J82, Linc00319 was overexpressed and promoted BLCA cell proliferation and invasion by sponging miR-4492 and regulating its target gene reactive oxygen species modulator 1 (ROMO1). Similar findings were obtained in another study. Wang et al. found that miR-3127 was downregulated in BLCA tissues and cell lines and directly interacted with RAP2A (a member of the small GTPase superfamily), thus inhibiting tumour growth and metastasis. However, linc00319 sponges miR-3127 and promote the proliferation and invasion of BLCA cells [81,82].

Linc00355 is located on chromosome 13q21.31 and has been validated to function as a ceRNA by sponging miRNAs in lung squamous cell carcinoma, glioma and hepatocellular carcinoma [83,84,85]. In BLCA, linc00355 could act as a sponge of miR-424-5p to modulate high mobility group AT-hook 2 (HMGA2) expression, which could regulate the EMT-related proteins ZEB1, E-cadherin and vimentin and finally contribute to BLCA EMT and lung metastasis [86]. In addition, studies by Yan et al. showed that linc00355 was highly expressed in cancer-associated fibroblast exosomes, and exosome-mediated linc00355 transfer promotes BLCA cell proliferation, invasion and chemotherapy resistance via miR-34b-5p/ATP binding cassette subfamily B member (ABCB1) and the miR-15a-5p/HMGA2 axis [87,88,89]. Moreover, based on qRT—PCR results and clinical data from 59 bladder cancer patients, linc00355 was elevated in BLCA tissues, especially in advanced bladder tumours, and was positively correlated with poor OS and recurrence-free survival (RFS) [86].

Linc00641 is located on 14q11.2 and is differentially expressed in a variety of cancers, such as non-small cell lung cancer, renal cell cancer, prostate cancer and bladder cancer [90,91,92,93]. Although linc00641 is upregulated in rectal cancer, gastric cancer and renal cell carcinoma, it is expressed at low levels in BLCA [90]. Patients with lower linc00641 expression levels had poorer OS and RFS. An overexpression experiment also proved that linc00641 is a tumour suppressor gene in bladder cancer. The BLCA cell lines T24 and J82 performed worse in proliferation, migration and invasion when linc00641 was overexpressed in vitro. A xenograft assay in nude mice showed the same result in vivo [93]. Further study demonstrated that linc00641 acts as a ceRNA of KLF10 by sequestering miR-197-3p. Upregulated KLF10 inhibits the PTEN/PI3K/AKT signalling pathway, thus decreasing the EMT of cancer cells [93].

Linc00649 is located on 21q22.11 and is widely considered an oncogene in multiple tumours, such as lung squamous cell carcinoma, breast cancer and gastric cancer [94,95,96]. A recent study found that linc00649 is a basement membrane-related lncRNA and is correlated with clinical prognosis based on analysis of transcriptional and clinical data of bladder cancer from the TCGA, GEO and BM-BASE databases, revealing that linc00649 is a potential biomarker of BLCA. Additionally, a model containing eight lncRNAs, including linc00649, was constructed and used to accurately predict the prognosis of BLCA patients [97]. Similar to other lincRNAs, linc00649 functions as a ceRNA of miRNAs at the transcriptional level. In the BLCA cell lines T24 and UM-UC-3, knockdown of linc00649 inhibited EMT in tumour cells in an HMGA1-dependent manner by sponging miR-15a-5p [98]. Another study also found that the sponge function of linc00649 led to a low expression level of the cancer suppressor gene miR-16-5p in BLCA cell lines, thus enhancing transcription of the miR-16-5p target gene Jumonji AT-rich interactive domain 2 (JARID2) [99].

Linc00958 is located on 11p15.3 and was found to be substantially expressed in bladder cancer tissues compared to normal bladder epithelial tissues. Increasingly, studies have demonstrated that linc00958 is involved in the malignant progression of various cancers, such as hepatocellular carcinoma, colorectal cancer, osteosarcoma and endometrial cancer [100]. Anna et al. identified 72 BLCA tissues and 8 normal bladder epithelial tissues by RNA sequencing and selected five significantly dysregulated lncRNAs, including linc00958, for further analysis. They found that the knockdown of linc00958 led to a loss of cell mobility in vitro. Moreover, RNA pull-down and RNA binding protein immunoprecipitation experiments indicated that metadherin (MTDH) might be the protein partner of linc00958 [101]. Another study screened the online database GEPIA and found that linc00958 was upregulated in all urinary tumours, including bladder cancer. However, the expression level of linc00958 was not necessarily correlated with OS and RFS in BLCA [102]. Other studies noted that linc00958 also regulates tumour growth, metastasis, angiogenesis and oxidative stress in BLCA by interacting with miRNAs or sponging them [103,104,105].

**Table 2 cancers-15-01007-t002:** The role of lincRNAs in bladder cancer.

Name	Expression	Related miRNAs, Proteins or Pathways	Related Tumour Cell Biology	Relevant Clinical Features	Reference
Linc00001 (XIST)	Upregulated	miR-200c, miR-355, miR-133a, TET1	Proliferation, migration, invasion, apoptosis and EMT	Tumour stage, metastasis, and prognosis	[51,52,53,54]
Linc00023 (MEG3)	Downregulated	miR-96, miR-27a, miR-494, p53	Proliferation, migration, invasion, apoptosis and chemosensitivity	Tumour stage, invasion and metastasis	[57,58,59,60]
Linc00047 (MALAT1)	Upregulated	miR-384, miR-124, miR-101-3p	Proliferation, migration, invasion and chemosensitivity	Tumour stage, invasion and prognosis	[63,64,65]
Linc00080 (TUG1)	Upregulated	miR-140-3p, miR-194-5p, miR-142, miR-145, HMGB1	Proliferation, migration, invasion, apoptosis, EMT, radiosensitivity and chemosensitivity	Tumour stage, invasion, LN metastasis and OS	[67,68,69,70,71]
Linc00178 (UCA1)	Upregulated	miR-195, miR-582-5p, miR-143	Proliferation, migration, invasion, autophagy, EMT, mitochondrial function and drug resistance	Tumour stage	[73,74,75,76]
Linc00319	Upregulated	miR-3127, miR-4492	Proliferation, migration and invasion	Tumour stage and RFS	[81,82]
Linc00355	Upregulated	miR-424-5p, miR-34b-5p, miR-15a-5p	Proliferation, migration, invasion, EMT and cisplatin resistance	Prognosis	[86,88,89]
Linc00641	Downregulated	miR-197-3p	Proliferation, migration and invasion	Prognosis	[93]
Linc00649	Upregulated	miR-15a-5p, miR-16-5p	Proliferation, migration and invasion	Muscle invasion and OS	[98,99]
Linc00958	Upregulated	miR-490-3p, miR-625-5p, miR-378a-3p, MTDH, SAPK	Proliferation, migration, invasion, apoptosis, autophagy and angiogenesis	OS	[101,102,103,104,105]

Abbreviations: XIST, X-inactive specific transcript; TET1, tet methylcytosine dioxygenase 1; EMT, epithelial-mesenchymal transition; MEG3, maternally expressed 3 gene; MALAT1, metastasis-associated lung adenocarcinoma transcript 1; PTEN, phosphatase and tensin homolog; FTO, fat mass and obesity-associated protein; MAL2, mal T-cell differentiation protein 2; TUG1, taurine-upregulated gene 1; HMGB1, high mobility group box 1 protein; LN, lymph node; OS, overall survival; UCA1, urothelial cancer associated 1; RFS, recurrence-free survival; MTDH, metadherin; SAPK, mitogen-activated protein kinase 9.

## 6. Conclusions

In this review, we summarized the functions and mechanisms of novel snoRNAs and lincRNAs in bladder cancer. Most studies have indicated that lincRNAs and snoRNAs are upregulated in bladder cancer and associated with oncogenesis, progression and metastasis, although GAS-5, MEG3 and linc00641 were shown to be tumour suppressor genes and were downregulated in BLCA. Their dysregulation in bladder cancer enables their use in the diagnosis and prognosis of BLCA. As shown in the tables, SNHG1, GAS-5, SNHG3, SNHG6, SNHG7, SNHG14 and SNHG16 are able to act as ceRNAs, sponge-specific miRNAs, stabilize target mRNAs and finally activate or inhibit important signalling pathways such as the Wnt/β-catenin, Src/FAK and PI3K/AKT pathways. In addition, SNHG1, GAS-5 and SNHG16 also directly bind to the histone methyltransferase EZH2 to influence the methylation of target genes and regulate cellular processes at the transcriptional level. LincRNAs also contribute to the progression of BLCA by sponging miRNAs. Interestingly, exosomes were found to participate in the transfer of linc00178 and linc00355, which indicates a potential mechanism of BLCA metastasis. In conclusion, emerging studies have focused on the role of snoRNAs and lincRNAs, revealing that snoRNAs and lincRNAs closely influence the development, progression and metastasis of BLCA in vivo and in vitro. With a better understanding of the specific mechanisms of snoRNA and lincRNA roles in BLCA, they show potential as biomarkers and therapeutic targets in the future.

## Figures and Tables

**Figure 1 cancers-15-01007-f001:**
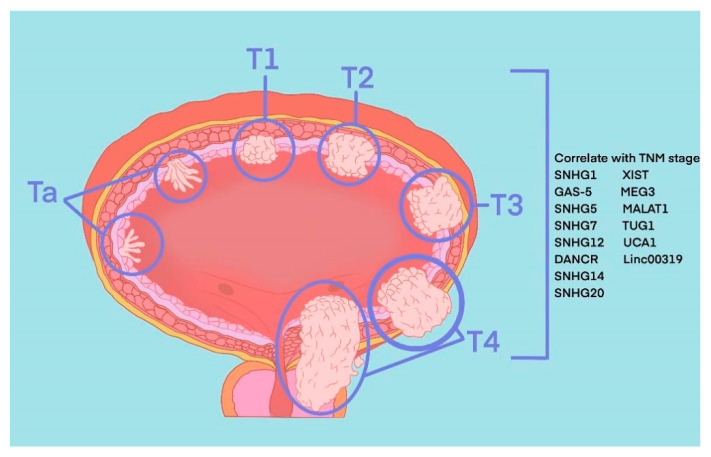
SNHGs and lincRNAs that correlate with TNM stage of BLCA.

**Figure 2 cancers-15-01007-f002:**
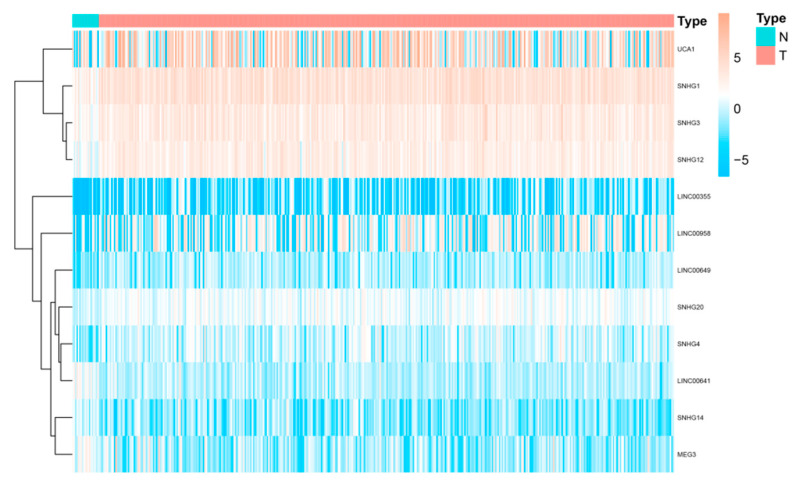
Differentially expressed SNHGs and lincRNAs between BLCA and paired normal samples.

**Figure 3 cancers-15-01007-f003:**
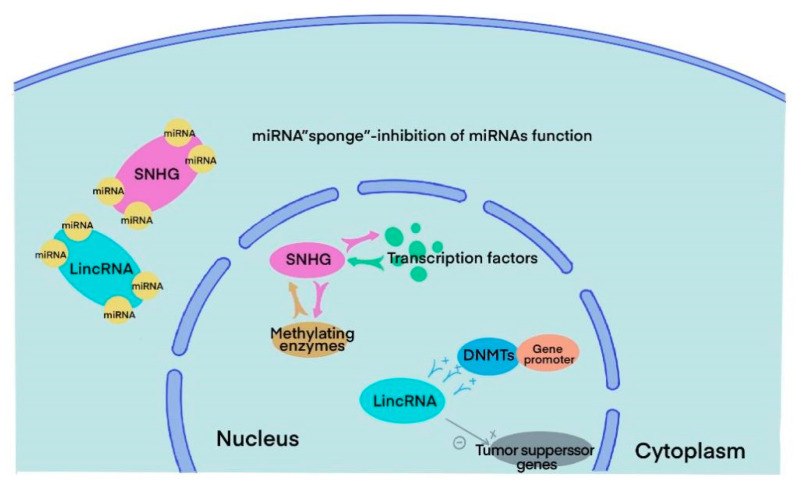
Major functions of SNHGs and lincRNAs in BLCA cells.

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
