# Peer review of "The Diagnostic and Therapeutic Role of snoRNA and lincRNA in Bladder Cancer"

_cancers, 2023, doi:10.3390/cancers15041007_

Round 1

Reviewer 1 Report

This paper is a comprehensive review of biological functions of snoRNA and lincRNA in bladder cancer. It lists a lot of snoRNAs and lincRNAs, introduced their basic information, and illustrated their functions in bladder cancer. The review is thorough, and relevant references are provided. It can be published after some minor revisions.

1. The authors should clarify the criteria for selecting these RNAs. Did they summarized all relevant literatures published recently?

2. Can the authors do a simple experiment on a dataset to simply illustrate the functions of these RNAs? For this aim, a heatmap or a differential expression analysis can be included in the paper.

Reviewer 2 Report

In this work, the authors investigated the role of snoRNAs and lincRNAs, revealing that snoRNAs and lincRNAs closely influence the development, progression, and metastasis of BLCA in vivo and in vitro. With a better understanding of the specific mechanisms of snoRNA and lincRNA roles in BLCA, they show potential as biomarkers and therapeutic targets in the future. I have gone through the manuscript, and I found the topic and the work done of good interest, and suitable for publication in “Caners”. The work presented is diversified and includes many important results. I recommended the manuscript for publication in “Caners”

Author Response

Thank you very much for your comments!

Reviewer 3 Report

The manuscript (cancers-2131061) entitled "The diagnostic and therapeutic role of snoRNA and lincRNA in bladder cancer" provided interesting information, although the organization of the manuscript is poor. The present form of the manuscript required extensive revision for its suitability for publication in cancers. 

1. It is suggested to include the pathophysiology of bladder cancer and correlate the role of snoRNA and lincRNA. This section should be enriched with good image illustrations for quick understanding to the reader.

2. It is suggested to signify the diagnostic and therapeutic role of snoRNA and lincRNA in a separate section. This section should be enriched with good image illustrations for quick understanding to the reader.

3. It is suggested to compile a separate table for the diagnostic role and therapeutic role of snoRNA in bladder cancer highlighting the outcomes of the related contemporary research. 

4. 3. It is suggested to compile a separate table for the diagnostic role and therapeutic role of lincRNA in bladder cancer highlighting the outcomes of the related contemporary research. 

5. Typo mistakes should be rectified throughout the manuscript including consistency in abbreviation. For example, Bladder cancer as BLCA, BCa 

Round 2

Reviewer 3 Report

The revised manuscript improved well, although the following minor issue should be addressed before publication.

1. Clarity of Table 1 is very hard to read; In my opinion, it should be split into two tables, or columns related to obvious information should be deleted. 

2. Clarity of Table 2 is very hard to read; In my opinion, it should be split into two tables, or columns related to obvious information should be deleted. 

3. Captions of Table 1 and 2 should be above of table, In the present manuscript, the table contains two captions, one above and the second below the table. This should be rectified. 
